# Effects of a Combined Intradialytic Exercise Training Program on Functional Capacity and Body Composition in Kidney Transplant Candidates

**DOI:** 10.3390/jfmk8010009

**Published:** 2023-01-11

**Authors:** Vasiliki Michou, Michaela Davioti, Niki Syrakou, Vasilios Liakopoulos, Asterios Deligiannis, Evangelia Kouidi

**Affiliations:** 1Sports Medicine Laboratory, School of Physical Education and Sport Science, Aristotle University, PC 57001 Thessaloniki, Greece; 2Division of Nephrology and Hypertension, 1st Department of Internal Medicine, Medical School, AHEPA Hospital, Aristotle University, PC 54636 Thessaloniki, Greece

**Keywords:** kidney transplant candidates, hemodialysis, intradialytic exercise, functional capacity, body composition, bioelectrical impedance analysis

## Abstract

Chronic kidney disease (CKD) leads to gradual muscle mass loss, which is strongly associated with lower functional capacity, which limits a patient’s daily activities. The aim of the present study is to examine the effects of a 4-month intradialytic exercise program on the functional capacity and body composition of kidney transplant (KT) candidates. Twenty-nine male patients on hemodialysis (HD) waiting for a kidney transplant, with a mean age of 53.86 ± 9.56 years old and BMI 27.11 ± 5.55 kg/m^2^, were randomly assigned into the following two groups: A (n_A_ = 15 HD patients), who followed a 4-month intradialytic exercise program combining aerobic and resistance training, with a supervised, progressively increasing workload, and B (n_B_ = 14 HD patients), who continued to receive usual care. At baseline and the end of the study, the KT candidates underwent a 6-min walking distance (6-MWD), and a 10-repetition sit-to-stand test (10-STS) to access physical function, a handgrip strength (HGS) test to evaluate the muscle strength of the non-fistula hand. Moreover, the bioelectrical impedance analysis (BIA) was performed to assess body composition indices, such as body fat (BF), body fat mass index (BFMI), fat-free mass index (FFMI), body cell mass (BCM), basal metabolic rate (BMR), extracellular water (ECW), intracellular water (ICW), total body water (TBW) and phase angle (PhA). Following the exercise program, group A showed favorable improvements in HGS (from 26.59 ± 9.23 to 28.61 ± 9.58 kg, *p* < 0.05) and 6-MWD (from 427.07 ± 7.66 to 468.16 ± 11.39 m, *p* < 0.05). Intergroup results from 6-MWD showed a statistically significant difference (Δp = 0.04), at the end of the study. Moreover, group A results from BIA revealed a significant increase of BMR by 2.4% (*p* < 0.05), ECW by 3.6% (*p* = 0.01), ICW by 3.8% (*p* = 0.01), TBW by 4.1% (*p* = 0.01), lean mass by 2.7% (*p* = 0.01), and PhA by 13.3% (*p* = 0.04), while a reduction in BF by 5.0% (*p* = 0.01) and BFMI by 6.6% (*p* = 0.03) was also noticed. At the end of the study, group A showed statistical differences in BMR (Δp = 0.01), BMR/BW (Δp = 0.01), dry lean (Δ*p* = 0.01), and PhA (Δp = 0.03), compared to the group B. Linear regression analysis in group A after training showed positive correlations between HGS and both PhA (r = 0.52, *p* = 0.04) and FFMI (r = 0.64, *p* = 0.01), and a strong negative correlation between 6-MWT and BF (r = −0.61, *p* = 0.01). In conclusion, a 4-month intradialytic exercise program can enhance body composition and some physical parameters in HD patients awaiting kidney transplantation.

## 1. Introduction

Progression of chronic kidney disease (CKD) is strongly associated with severe complications, such as increased incidence of cardiovascular disease, hyperlipidemia, anemia, musculoskeletal disorders, fatigue, and cognitive impairment [1,2] Moreover, CKD patients experience multiple nutritional and catabolic alterations that are commonly referred to as protein–energy wasting (PEW) [3]. In addition, sarcopenia, decreased muscle strength, and functional deterioration are correlated with PEW in CKD patients [4]). CKD patients have very low levels of daily physical activity (DPA), adopting a sedentary lifestyle, which eliminates their functional capacity, leads to muscle mass loss, and increases morbidity and mortality [5].

Physically inactive CKD patients experience a continuous decline in physical activity status, up to 3.4% each month after starting dialysis therapy. Maintenance hemodialysis (HD) is the most common renal replacement treatment for patients with CKD [6] while waiting for a kidney transplant (KT). Inactive HD patients have a 62% greater risk of 1-year mortality than those more physically active [7,8]. Some of the factors that contribute to a lower physical performance are the CKDs progression, the side effects of the hemodialysis treatment, fatigue, and comorbidities [7]. A 6-min walking test (6-MWT) and a sit-to-stand test (STS) are useful and validated tools for assessing exercise capacity and total physical performance in CKD patients [9,10]. Due to the necessity of a tool that can screen malnutrition and assess muscle strength and frailty, the handgrip strength (HGS) test has been suggested. HGS could be a valuable tool for assessing frailty and maximum isometric muscle strength related to prognosis in HD patients [11]. It has also been suggested as a biomarker linked with nutritional decline [12,13] and mortality in HD patients [14,15]. Studies have also found correlations between muscle function and the prediction of CKD complications [16]. In a recent study, physical performance, as it was measured with functional tests, such as the 6-min walking test (6-MWT) or sit-to-stand test (STS), was highly associated with higher HGS values [17].

Systematic exercise training has favorable effects both on the functional capacity and clinical outcomes in CKD patients undergoing HD. According to the Renal Association Guidance [18], HD patients should be encouraged to increase their physical activity and participate in structured interventional exercise programs. More specifically, HD patients should exercise at least 150 min throughout the week at a moderate intensity. Nowadays, most exercise training programs are implemented during the HD sessions, although supervised exercise programs are also applied at home and in rehab centers on the non-dialysis days. It is a fact that HD patients attend dialysis units three times per week and spend each session in a sitting position or lying on a bed for almost 4 h. Not surprisingly, several studies demonstrated that intradialytic exercise training is associated with higher adherence rates, compared to the one performed on the non-dialysis days [19,20]. The literatures results have shown remarkable improvements in 6-MWT and STS after long-term intradialytic exercise [21,22,23]. However, studies often come up with controversial results regarding body composition evaluation after an aerobic or resistance exercise training session. Normal or higher values of free-fat mass index (FFMI) and BMI (body mass index) are considered to improve survival in HD patients, while increased muscle mass is strongly associated with a better functional capacity [24]. Conversely, a significant BMI reduction is not always desirable for HD patients, as is often hard to identify if the lower BMI values are associated with a decrease in fat mass (FM) or lean body mass (LBM) [25].

This randomized controlled trial (RCT) aimed to evaluate the effects of a 4-month combined intradialytic exercise training on functional capacity and body composition indices in KT candidates.

## 2. Methodology

### 2.1. Patients

KT candidates undergoing maintenance HD were recruited from the Dialysis Unit of the AHEPA University Hospital, Thessaloniki, Greece and were screened for eligibility. Patients, who met the inclusion criteria and volunteered to participate in our study, were randomly assigned to either group A or B. All patient’s inclusion criteria included as follows: being on the waiting list for KT, no history of coronary heart disease within the previous six months, and no severe musculoskeletal problems that may limit patient’s participation in the study. Patients with unstable hypertension, unstable angina, severe anemia (Ht < 25% or Hb < 8.5 g/dL), lack of compliance with medications and hemodialysis treatment, and concurrent involvement in a similar exercise training program, were excluded.

### 2.2. Study Design

Patients who met the inclusion criteria and volunteered to participate in this clinical trial underwent a clinical examination, a 6-min walking test (6-MWT), and 10-repetition sit-to-stand test (10-STS) to assess functional capacity, a handgrip strength (HGS) test to evaluate the maximum isometric strength of hand and bioelectrical impedance analysis to estimate body composition parameters. Following the above measurements, patients were randomly assigned to either an exercise (group A) or control (group B) group, by the www.randomizer.org website (assessed on 28 February 2022). Patients in group A followed a 4-month intradialytic aerobic with a static bike and strength with TheraBands and dumbbells exercise training program, while group B remained untrained. During the study period patients’ dialysis method and medication remained unchanged to exclude pharmacological side effects. The protocol of the study was approved by the Ethics Committee of the Department of Physical Education and Sports Science of Aristotle University of Thessaloniki (Protocol number: 121/2022). Before enrollment, KT candidates provided written informed consent before enrollment. The clinical trial started in March of 2022 and ended in June 2022.

### 2.3. Anthropometric Measurements

At baseline and after the end of the study, after the HD session, the SECA electronic scale was used to measure dry weight (kg) and height (cm).

### 2.4. Six-Minute Walking Test

Six-minute walking test (6-MWT), which is a validated assessment tool for the exercise capacity in HD patients [9,26], was conducted in both groups on a non-dialysis day. Before the test, patients were instructed to wear comfortable clothing and shoes and take their medication as usual. They also received specific instructions about how to perform the test. Then, a few hours before the test, a trundle wheel was used to measure the track of the test, while a piece of tape was used to mark the end of the walkway. At the beginning of the test, there was a 1-min warm-up period. After the warm-up section, patients were asked to walk up and down a 30-m hospital corridor without assistance for 6 min. During the test, patients were continuously provided with encouragement in specific even tones. The researcher used a timer or a stopwatch to measure the test duration. The distance in meters, which each patient covered during the 6 min (6-MWD) was recorded in the data collection analysis form.

### 2.5. 10-Repetition Sit-to-Stand Test

The 10-repetition sit-to-stand test (10-STS) was also used to accesses physical function in our KT candidates. Patients were instructed to sit on a regular steel chair (45 cm height and 61cm depth) [27], which was not adjusted for every patient, with their feet touching the floor, their knee joint at 90 degrees, their back touching the chair’s back, and their arms crossed on the shoulders at chest height. On the “go” command, patients were asked to perform ten consecutive repetitions of sitting down and getting up from a chair, as fast as possible, in the waiting room of the dialysis unit. The test was performed once at baseline and once at the end of the study. Data was collected in time (seconds).

### 2.6. Handgrip Strength Test

HGS (Jamar^®^ Plus Digital Hand Dynamometer, Sammons Preston, Chicago Illinois, USA) was measured before the HD session from the dominant non-fistula arm for patients with arteriovenous fistula and the dominant hand for patients with a central venous catheter (CVC). At baseline, patients were asked to self-adjust the dynamometer to fit their hand size to obtain the best performance. Then in sitting position, our patients were instructed to extend the dominant hand in front of the body, in a transverse plane. The dynamometer handle was tightened with the patient’s elbow at a 90° angle and a short distance from the trunk. Each patient performed three maximum active muscle contraction trials, in response to a voice command, with an interval of 10 s. In each trial, patients were encouraged to grasp the dynamometer as hard as possible until the peak-hold stops increased. Then the researcher stopped the measurement and that was the duration of each muscle contraction. The maximal force of each grip effort was measured and the mean value of the 3 trials was used for results analysis. The total duration of the test was about 4–5 min.

### 2.7. Bioelectrical Impedance Analysis

The bioelectrical impedance analysis (BIA) was used to evaluate body composition parameters. One day before the test, patients received the following instructions: (a) be in a fasted state from food for at least 2 h and beverages for at least 12 h, (b) avoid any exercise training for at least 12 h, (c) inform the researcher for any possible medication that may affect body fluid and electrolyte balance [28]. Quadscan 4000 machine (Bodystat, Warwickshire, UK), was used to assess body composition indices, 30 min after the end of an HD session, on a dialysis day. Patients were instructed to remain supine, with their limbs not touching their torso, and to remove any metal objects touching their skin. Two pairs of disposable electrodes were placed on patients’ limbs. One pair was placed on the dorsum of the hand over the third metacarpophalangeal joint and the wrist. In contrast, the other was placed over the ipsilateral third metatarsophalangeal and ankle joints. The body fat (BF), body fat mass index (BFMI), fat-free mass index (FFMI), body cell mass (BCM), basal metabolic rate (BMR), ratio of basal metabolic rate and body weight (BMR/BW), extracellular water (ECW), intracellular water (ICW), total body water (TBW), and phase angle (PhA), were assessed. BFMI and FFMI were automatically calculated by the Quadscan 4000 machine. However, it is well known that BFMI can be calculated by dividing BF (kg) by the height squared (m^2^) and FFMI can be estimated by dividing free-fat mass (FFM) (kg) by the square of height (m^2^) [29].

### 2.8. Intradialytic Exercise Program

Group A followed a 4-month combined intradialytic exercise training, three times a week, with 80–100 min duration during the HD sessions in the dialysis unit. Exercise intensity was set at 13–14 (somewhat hard), according to the Borg’s Rating of Perceived Exertion (RPE) scale. Specialized exercise trainers at physical rehabilitation conducted each exercise session, starting during the first 2 h of each HD session. The exercise training divided into the following 4 parts: a 5-min warm-up, aerobic exercise training for 40–60 min by using stationary bicycles (Moto Med Letto 713/w1498, Tampa, FL, USA), which were adjusted to each patient’s bed or chair and patients used their strength against the bicycle resistance (active setting) and strength training for approximately 20 min, which was divided into 3 sets of 12–15 repetitions (with a 45–60” second passive break between the sets) by using resistance TheraBands, dumbbells (1 kg) and stress balls and a 5-min recovery (Figure 1). According to each patient’s ability, the duration of cycling was gradually increased over time and finally reached 60 min of active cycling.

For safety reasons, heart rate was continuously monitored during each exercise session by a pulse oximeter sensor, while blood pressure was measured every 15 min.

### 2.9. Statistical Analysis

For the statistical analysis, the IBM Statistical Package for Social Sciences (IBM Corp. Released 2020. IBM SPSS Statistics for Windows, Version 27.0. Armonk, NY, USA: IBM Corp.) was used. By using the Kolmogorov–Smirnov test the normal distribution of variables was assessed. A two-way ANOVA with repeated measures was applied to evaluate mean differences within time and between the two groups. The differences found between groups A and B in terms of changes in the examined parameters were analyzed with the t-test for independent samples. Linear regression was also used to find the association between variables that revealed statistically significant changes over time. Moreover, for normally distributed variables, data were refed as mean ± standard deviation. The effect size was used to calculate the relationship strength between the two groups variables. A value of less than 0.2 refers to a small effect size, 0.4 refers to a median size effect, and over 0.8 refers to a high effect size. The difference score (Δ) between a pre- and a post-test score in all variables was estimated. The significance level for accepting or not having a statistically significant difference obtained for all statistical tests was set at *p* < 0.05.

## 3. Results

Initially, 50 KT candidates were screened for eligibility. Forty of these patients, who met the inclusion criteria and volunteered to participate in our study, were randomly assigned to either group A or B. Five patients from group A and six from group B withdrew during the follow-up period; therefore, 29 patients completed the study (Figure 2). During the 4-month period, none of the patients showed any exercise-induced cardiovascular or musculoskeletal complications. The clinical characteristics of the KT candidates are shown in Table 1.

At baseline, there were no statistically significant differences in functional capacity measurements, HGS test, and BIA between groups A and B. HGS was performed with the right arm in 86.6% of patients in group A and 78.6% in group B. After the 4-month intradialytic training program, group A showed a significant increase of HGS by 7.5% (*p* < 0.05), and 6-MWD by 9.6% (*p* < 0.05), while 10-STS time performance did not statistically improve. In addition, inter-group results from 6-MWD showed a statistically significant difference (Δp = 0.04), after 4 months (Table 2).

In addition, regarding the effects of exercise on BIA, results showed that after 4 months, group A showed higher values in BMR by 2.4% (*p* < 0.05), ECW by 3.6% (*p* = 0.01), ICW by 3.8% (*p* = 0.01), TBW by 4.1% (*p* = 0.01), lean mass by 2.7% (*p* = 0.01), and PhA by 13.3% (*p* = 0.04), while lower values were observed in BF by 5.0% (*p* = 0.01) and BFMI by 6.6% (*p* = 0.03). At the end of the study, group A showed statistical differences in BMR (Δp = 0.01), BMR/BW (Δp = 0.01), dry lean (Δp = 0.01), and PhA (Δp = 0.03), compared to the group B (Table 3).

Lastly, at the end of the study a positive linear relationship was found in group A between HGS and PhA (*r* = 0.52, *p* = 0.04) (Figure 3), HGS and FFMI (*r* = 0.64, *p* = 0.01) (Figure 4), while a negative linear relationship was noticed only between 6-MWD and BF (*r* = −0.61, *p* = 0.01) (Figure 5).

## 4. Discussion

The present study demonstrated that a 4-month combined intradialytic exercise training had favorable effects on the functional capacity and body composition of KT candidates. It is well-known that CKD leads to muscle mass loss and deterioration of functional capacity, which usually causes loss of the ability to perform activities of daily living. This phenomenon is much worse for elderly HD patients, as the prevalence of sarcopenia ranges at higher levels [30]. Moreover, previous studies have shown that reduced exercise capacity in HD patients is strongly associated with significant muscle atrophy, mainly type II, and a higher decrease in muscle strength [31].

For decades, systematic exercise training programs in CKD patients have shown favorable effects on physical function, aerobic capacity, body composition, cardiovascular morbidity, mental health, and quality of life [32]. Intradialytic exercise training is an easy and well-tolerated exercise method that encourages patients to be physically active and increases their compliance [19,33]. Previous studies have shown that a long-term intradialytic exercise program, 2 or 3 times per week may decrease fatigue, subsequently enhancing exercise tolerance and leading to favorable cardiovascular and muscular adaptations [34]. This study revealed a significant improvement in physical performance (as measured with 6-MWT and 10-STS), muscle strength, and body composition indices in HD patients after a 4-month combined intradialytic exercise training.

Our study revealed a significant improvement in 6-MWD by 9.6% in our exercise KT candidates. The distance walked in the 6-MWT is an independent mortality factor for end-stage kidney disease patients undergoing hemodialysis. Several studies have previously shown that 6-MWT is a low-cost and applicable tool for functional capacity evaluation [35]. Even though many studies have previously evaluated patients’ physical performance with 6-MWT, the results are controversial. In a meta-analysis, Heiwe and Jacobson [36] reported that exercise programs that differed in type, frequency, and duration did not improve the distance walking tests overall. In contrast, in a pilot study, Desai et al. [20] showed that after a 4-month aerobic intradialytic exercise program, 6-MWD was statistically increased by 12.3%. Similarly, Torres et al. [37] and Frih et al. [38] by applying a similar aerobic exercise program in HD patients, noted a significant improvement in 6-MWT distance by 6.5% and 15.9%, respectively. Furthermore, Abdelaal and Abdulaziz [39] investigated the effect of a 3-month intradialytic cycling program and a 3-month resistance training program with a 2-month post-study cessation. They found that after the end of the study, the aerobic group had a 33.1% increase in the distance in 6-MWT, while the resistance training group increased performance in 6-MWT by 22.5%.

Different types of sit-to-stand tests can also assess physical performance. According to Segura-Ortı [10], 10-STS, 60-s-STS, and 6-MWT have high test-retest reliability in HD patients. So, combining these tools in monitoring physical performance changes and assessing the effectiveness of different exercise training programs is clinically valuable for the HD population [10]. Rhee et al. [23] showed that after a 6-month intradialytic combined exercise program, three times per week, patients increased the number of repetitions in the 60-s-STS by 38.2%. At the same time, a significant increase of 6-MWD by 11.3% was also noticed. In addition, Hellberg et al. [22] by comparing the effects of a 12-month balance and strength exercise in 151 HD patients, found that both groups statistically improved physical performance. More precisely, the strength group showed a favorable increase of 6-MWD by 17.8% and 30-s-STS by 15.4%, while the balance group showed an increase of 9.6% in 6-MWD and an increase of 7.7% in 30-s-STS. Similarly, Frih et al. [38] showed that after a 4-month combined endurance-resistance training program on the non-dialysis days, 10-STS was statistically increased by 16.2%. Results from our study did not show a significant improvement within group A. On the contrary, the 10-STS performance had significantly deteriorated in the exercise group and considerably improved in the controls. Moreover, intergroup results for 10-STS showed that the control group had a favorable improvement in time compared to group A. However, there is a discussion about the STS test reliability and factors that may affect these results. Previous studies have revealed that age and HD vintage are predictors of worse physical functioning, including sit-to-stand tests. In the study of Kim et al. [5], STS showed unsatisfactory results for HD adult patients and did not significantly correlate with daily physical activity, compared to 6-MWD. While in the study of Santos et al. [40] age and duration of HD were found to affect the 60-s-STS performance.

Moreover, the present study revealed a statistically significant increase in HGS (by 7.5%) for the exercise group after training. HGS is considered a valuable clinical and functional measure that can define sarcopenia, malnutrition, or frailty [41,42]. The HGS measurement has also been used in HD patients, and, by specific cut-offs (e.g., 28.3 kg for males and 23.4 kg for females), can predict both malnutrition and mortality [43,44]. In a recent meta-analysis, Hwang et al. [11] showed that a low HGS is strongly correlated with an increased risk of all-cause mortality in CKD patients undergoing HD. Similarly, Matsuzawa [45], in a prospective cohort study of 817 HD patients, provided evidence that linked decreased functional capacity and high mortality risk.

Systematic and long-term exercise can improve HGS levels, as well as body composition indices. Our study revealed significant improvements both for HGS and body composition indices. These findings are of high importance, as PEW or frailty among HD patients has been related to increased cardiovascular risk and mortality [46]. Several studies have strongly correlated higher HGS levels with higher muscle mass and overall survival in CKD patients [11,47,48]. In the study of Desai et al. [20] and Frih et al. [38], HGS was statistically increased both after intradialytic and non-intradialytic exercise training. While Suhardjono et al. [49] by dividing 123 HD patients into aerobic, combined, and control groups, found that after 12 weeks of twice-a-week intradialytic exercise training, there was no difference in overall outcomes for HGS and body composition parameters between the aerobic and combined group.

BIA is a safe, validated, and non-invasive method to evaluate body composition indices, as well as a currently increasingly used tool for dry weight and nutritional status management in HD patients [50]. In a recent systematic review, Bakaloudi et al. [25] showed that BIA, BCM, and DEXA are among the most widely used body composition assessment tools in interventional exercise programs for HD patients. However, according to Pajek et al. [51], the discrepancy between different body composition devices that may affect measurements in HD patients should be considered. For example, they found that the BIS BCM Fresenius device and the MF-BIA InBody 720 yielded significant differences in TBW, ICW, BF, and BCM indices. Similarly, Jaffrin and Morel [52] and Mager et al. [53] also noticed different results in healthy and patient individuals by comparing BIS and BIA.

Our study revealed significant improvements within group A in BF, BFMI, FFMI, BMR, and TBW, as well as significant differences in 6-MWD, BMR, BMR/BW, dry lean, and PhA between groups at the end of the study. In addition, a strong correlation between HGS and FFMI and 6-MWD and BF was also noticed. Similarly, Torres et al. [37] found that after a 3-month intradialytic exercise training at moderate intensity, BMI, LTI (lean tissue index), and ICW were statistically improved for the exercise group compared to the control group. In contrast, Vogiatzaki et al. [54] showed that after a 6-month of intradialytic cycling training, there were no significant changes in any body composition indices. Furthermore, Moraes et al. [55] showed that a 6-month resistance training program can statistically increase FFMI, while the fat mass remained almost unchanged. While Masajtis-Zagajewska et al. [8] showed that BMI, TBW, ICW, and BCM were statistically decreased in non-dialysis patients with CKD at stage 3 or 4 after a 3-month aerobic exercise training 5 times per week at 60–80% of the maximum heart rate (HRmax). Similarly, Desai et al. [20] showed that none of the body composition indices assessed with BIA have changed after a 4-month aerobic intradialytic exercise program. While Dong et al. [56] observed that after a 12-week progressive intradialytic resistance cycling, there were no significant differences for the exercise group at the end of the study. The BMI and BFMI differences were significant only in comparison between groups 12 weeks later.

Among the above-mentioned body composition indices, PhA, the cell membrane integrity and vitality index, is the most clinically relevant impedance parameter [57]. PhA is a prognostic indicator of morbidity and mortality in various chronic conditions. It is also a valuable tool for assessing a disease’s progression [58]. A higher PhA indicates large quantities of intact cell membranes, while a lower PhA indicates cell death or reduced cell integrity [59]. Saitoh et al. [50], in a retrospective cohort study, showed that a lower PhA was strongly associated with a greater risk of PEW and frailty in HD patients. While Kang et al. [17] showed that PhA was associated with HGS, 5-STS, and 6-MWT performance. They also noticed that HD patients with higher values of PhA had favorable effects on the hospitalization-free survival rate and patient survival rate, and those with severe limitations in physical activity had lower values of PhA. In agreement with this study, Martin-Alemañy et al. [57] found a significant increase in PhA after a 3-month resistance intradialytic exercise training with or without oral nutritional supplementation for both the control and exercise group. These results indicate that exercise does not seem to augment the acute anabolic effects of intradialytic oral nutritional supplementation on nutritional status [57]. Similarly, Yuguero-Ortiz et al. [21] showed that after a 6-month combined intradialytic exercise training twice a week, even though significant increases were noticed in 6-MWD, HGS, 10-STS, and BMI, PhA did not statistically change. Our results showed a significant increase in PhA and a positive correlation between HGS and PhA at the end of the study in our exercised KT recipients. According to previous studies, this finding can be highly acknowledged as patients with higher physical activity levels can stay physically fit enough for transplantation and significantly improve morbidity and mortality.

Lastly, this study has strengths and limitations that need to be considered. Firstly, this study has revealed significant improvements in functional capacity, muscle strength, and body composition indices. It has also shown significant correlations at the end of the study that linked exercise’s value with physical performance and body composition indices. On the other hand, a limitation of this study was the small number of participants, mainly due to the difficulties in recruiting KT candidates that meet the inclusion criteria for long-term exercise training. Moreover, the fact that only males participated in our study, the small number of repetitions in the STS test, the 10-s interval between the HGS trials and the lack of patients’ physical activity history before the study may also be considered as limitations of the study.

## 5. Conclusions

In conclusion, a 4-month, intradialytic combined exercise training program can improve body composition and some physical parameters (HGS and 6-MWD) in the intervention group of HD patients while waiting for a kidney transplant. Thus, KT candidates should be encouraged to live an active life throughout each day to reach the optimal level of their functional and nutritional status.

## Figures and Tables

**Figure 1 jfmk-08-00009-f001:**
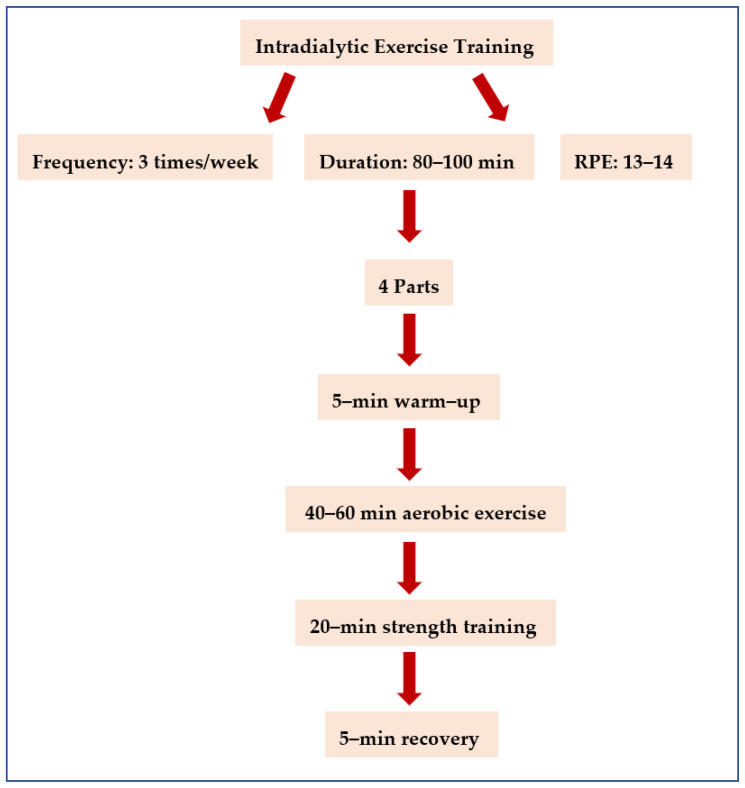
Flowchart of the intradialytic exercise program.

**Figure 2 jfmk-08-00009-f002:**
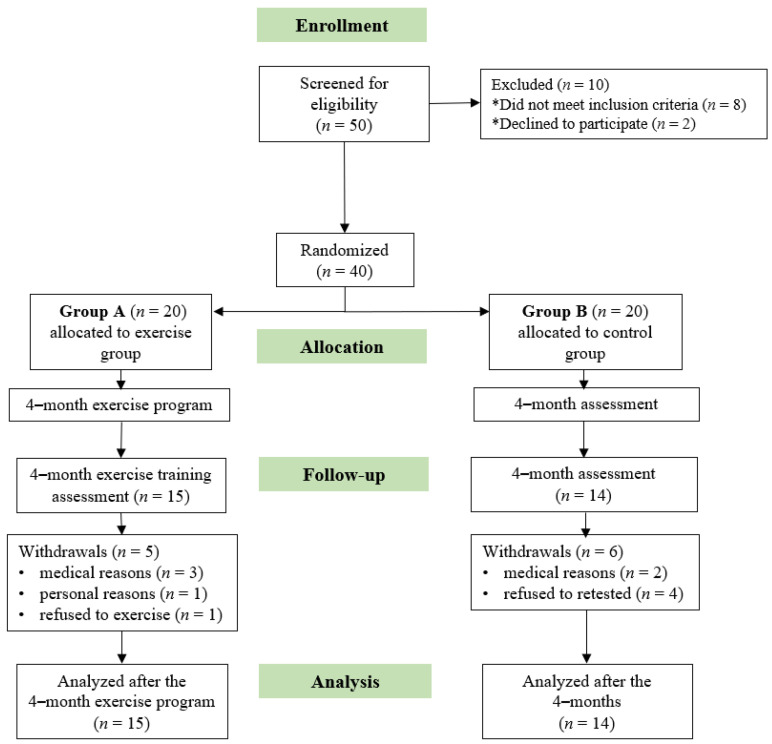
Flow chart diagram of the study.

**Figure 3 jfmk-08-00009-f003:**
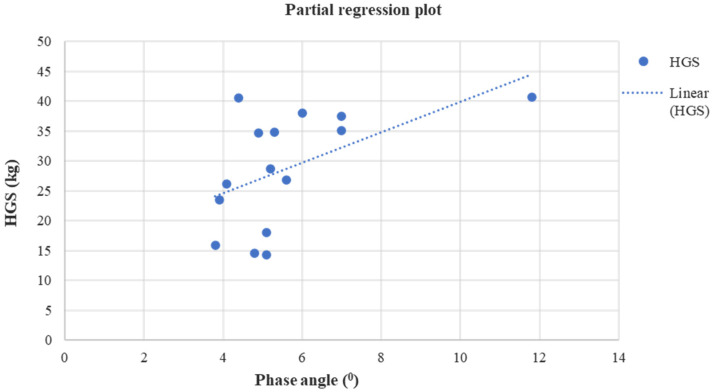
Linear regression analysis between the HGS (kg) and PhA (^0^) after four months in group A (*r* = 0.52, *p* = 0.04).

**Figure 4 jfmk-08-00009-f004:**
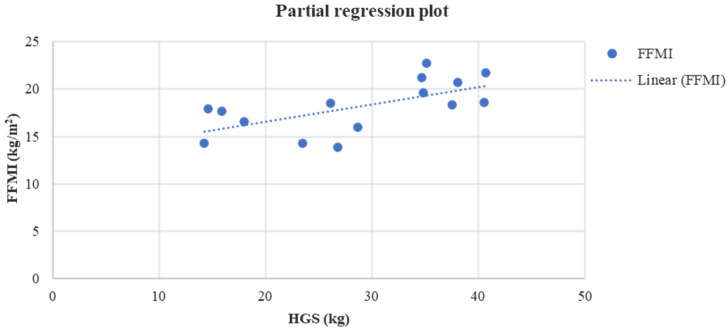
Linear regression analysis between the FFMI (kg/m^2^) and HGS (kg) after four months in group A (*r* = 0.64, *p* = 0.01).

**Figure 5 jfmk-08-00009-f005:**
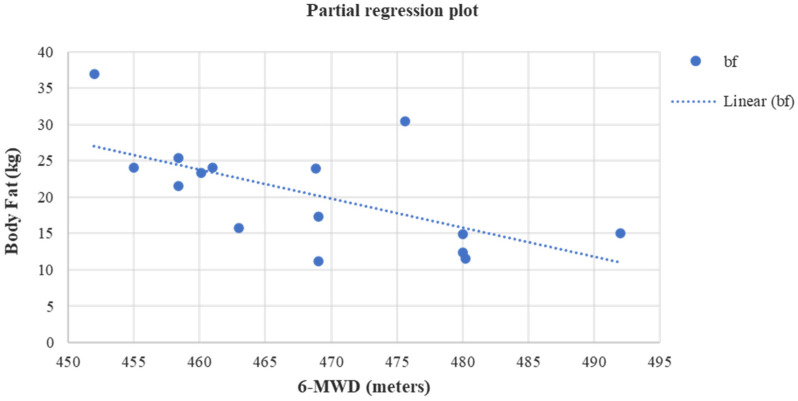
Linear regression analysis between the body fat (kg) and 6-MWD (meters) after four months in group A (*r* = −0.61, *p* = 0.01).

**Table 1 jfmk-08-00009-t001:** Clinical characteristics of the KT candidates.

	Group A	Group Β	*p*/ES
Number of participants	15	14	*p* = 0.87 (0.006)
Age (years)	53.26 ± 9.48	54.50 ± 9.95	*p* = 0.73 (0.05)
Height (cm)	1.70 ± 0.07	1.68 ± 0.08	*p* = 0.37 (0.45)
Dry weight (kg)	78.96 ± 17.56	80.47 ± 21.73	*p* = 0.67 (0.30)
BMI (kg/m^2^)	25.63 ± 4.19	28.69 ± 6.49	*p* = 0.42 (0.18)
HD vintage (years)	95.41 ± 77.22	96.67 ± 70.13	*p* = 0.77 (0.03)
Dialysis access			
− Arteriovenous fistula or graft	10	8	*p* = 0.45 (0.28)
− Central venous catheter	5	6	*p* = 0.87 (0.007)
Comorbidities			
− Diabetes mellitus	2	1	*p* = 0.89 (0.006)
− Hypertension	11	12	*p* = 0.91 (0.01)
− Dyslipidemia	3	2	*p* = 0.89 (0.009)
− Diabetic retinopathy	2	1	*p* = 0.92 (0.02)
− Osteoporosis	4	2	*p* = 0.76 (0.04)
− Chronic obstructive pulmonary disease (COPD)	1	0	*p* = 0.88 (0.008)
− Other	1	2	*p* = 0.81 (0.007)
Ht (%)	36.2 ± 4.0	35.9 ± 4.1	*p* = 0.55 (0.23)
Hb (mg/dL)	12.3 ± 1.2	12.3 ± 1.1	*p* = 0.81 (0.008)
Urea (mg/dL)	106.1 ± 30.9	107.0 ± 31.1	*p* = 0.65 (0.27)
Creatinine (mg/dL)	11.2 ± 2.5	11.4 ± 2.1	*p* = 0.94 (0.02)
URR (%)	74.9 ± 3.2	75.0 ± 3.4	*p* = 0.70 (0.03)
Kt/V	1.25 ± 0.0	1.27 ± 0.1	*p* = 0.58 (0.17)

Note: ES: effect size; *p*: *p*-value; BMI: body mass index; HD vintage: is the patients’ history (in years) in hemodialysis therapy; Ht: Hematocrit; Hb: Hemoglobin; URR: urea reduction ratio; Kt/V: is an index that assesses dialysis adequacy (K: urea clearance by dialysis, t: dialysis time, V: distribution urea volume); Independent t-test for continuous variables; Significant at the 0.05 level (*p* < 0.05).

**Table 2 jfmk-08-00009-t002:** Functional capacity analysis of the KT candidates at baseline and the end of the study.

	Group A				Group B				A vs. B Group			
	Baseline	After 4-Months	*p* (ES)	Δ (CI)	Baseline	After 4-Months	*p* (ES)	Δ (CI)	Pre *p* (ES)	Δp	Post *p* (ES)	Δp
Handgrip Strength (kg)	26.59 ± 9.23	28.61 ± 9.58	*p* < 0.05 (1.41)	−2.01 (1.25/2.89)	26.26 ± 6.55	26.00 ± 6.62	*p* = 0.08 (1.39)	−0.26 (−0.12/0.64)	*p* = 0.12 (0.01)	*p* = 0.32	*p* = 0.07 (0.03)	*p* = 0.32
10-STS (sec)	22.40 ± 5.50	24.80 ± 6.33	*p* < 0.05 (0.41)	2.40 (1.42/3.43)	22.07 ± 2.99	20.42 ± 3.73	*p* < 0.05 (1.73)	0.5 (−1.45/0.45)	*p* = 0.18 (0.02)	*p* = 0.46	*p* < 0.05 (0.14)	*p* = 0.05
6-MWD (m)	427.07 ± 7.66	468.16 ± 11.39	*p* < 0.05 (1.69)	−19.76 (−44.80/−2.39)	420.55 ± 13.17	408.54 ± 10.85	*p* = 0.16 (0.39)	12.01 (7.98/16.04)	*p* = 0.08 (0.09)	*p* = 0.51	*p* = 0.06 (0.10)	*p* = 0.04

Note: ES: effect size; *p*: *p*-value; Δ: average difference; CI: 95% confidence interval (lower bound/upper bound); STS: 10 times sit-to-stand test; 6-MWD: 6-min walking distance; Data are expressed as mean ± SD; *p* < 0.05: baseline vs. 6 months follow-up; *p* < 0.05: group A vs. B.

**Table 3 jfmk-08-00009-t003:** Bioelectrical impedance analysis of the KT recipients at baseline and the end of the study.

	Group A				Group B				A vs. B Group			
	Baseline	After 4-Months	*p* (ES)	Δ (CI)	Baseline	After 4-Months	*p* (ES)	Δ (CI)	Pre *p* (ES)	Δp	Post *p* (ES)	Δp
BF (kg)	21.61 ± 6.62	20.52 ± 7.39	*p* = 0.01 (0.52)	−1.09 (−2.40/0.04)	28.57 ± 12.21	28.82 ± 12.68	*p* = 0.51 (0.17)	0.26 (−0.59/1.12)	*p* = 0.15 (0.12)	*p* = 0.37	*p* < 0.05 (0.14)	*p* = 0.15
BFMI (kg/m^2^)	7.12 ± 2.09	6.65 ± 2.06	*p* = 0.03 (0.68)	−1.09 (−2.40/0.04)	9.41 ± 3.15	9.57 ± 3.38	*p* = 0.43 (0.27)	0.16 (−0.17/0.50)	*p* = 0.06 (0.16)	*p* = 0.09	*p* = 0.12 (0.22)	*p* = 0.07
FFMI (kg/m^2^)	18.54 ± 2.60	19.08 ± 2.63	*p* = 0.05 (0.60)	0.13 (0.08/1.10)	18.47 ± 2.94	18.47 ± 2.66	*p* = 0.51 (0.01)	0.007 (−0.33/0.35)	*p* = 0.87 (0.00)	*p* = 0.32	*p* = 0.85 (0.01)	*p* = 0.31
BCM (kg)	33.16 ± 6.51	34.10 ± 6.07	*p* = 0.11 (0.53)	1.39 (−0.11/1.99)	31.97 ± 7.30	31.09 ± 7.59	*p* = 0.96 (0.46)	−0.88 (−1.99/0.21)	*p* = 0.90 (0.008)	*p* = 0.30	*p* = 0.44 (0.04)	*p* = 0.26
BMR (kcal)	1713.80 ± 291.41	1754.53 ± 282.43	*p* < 0.05 (0.70)	40.73 (11.01/−76.41)	1583.07 ± 278.22	1571.21 ± 275.63	*p* = 0.57 (0.42)	−11.85 (−27.94/4.23)	*p* = 0.97 (0.05)	*p* = 0.06	*p* = 0.90 (0.10)	*p* = 0.01
BMR/BW (kcal/kg)	21.95 ± 1.77	22.47 ± 2.00	*p* = 0.06 (0.79)	0.52 (0.17/0.94)	20.13 ± 2.28	19.87 ± 2.27	*p* = 0.13 (0.41)	−0.25 (−0.61/0.10)	*p* = 0.47 (0.17)	*p* = 0.16	*p* = 0.54 (0.28)	*p* = 0.01
ECW (lt)	17.66 ± 2.65	18.30 ± 2.91	*p* = 0.01 (0.52)	0.64 (−0.03/1.41)	17.10 ± 2.91	17.57 ± 2.67	*p* = 0.15 (0.42)	0.47 (−0.18/1.14)	*p* = 0.94 (0.01)	*p* = 0.12	*p* = 0.45 (0.01)	*p* = 0.10
ICW (lt)	23.00 ± 4.15	23.87 ± 4.23	*p* = 0.01 (0.72)	0.87 (0.16/1.60)	22.39 ± 5.10	22.81 ± 4.86	*p* = 0.14 (0.45)	0.42 (−0.11/0.95)	*p* = 0.75 (0.05)	*p* = 0.15	*p* = 0.96 (0.01)	*p* = 0.12
TBW (lt)	41.10 ± 6.88	42.80 ± 7.00	*p* = 0.01 (0.68)	1.7 (0.37/3.28)	39.22 ± 7.20	40.03 ± 7.18	*p* = 0.11 (0.44)	0.81 (−0.05/1.68)	*p* = 0.84 (0.01)	*p* = 0.08	*p* = 0.70 (0.03)	*p* = 0.06
LEAN (kg)	56.96 ± 11.26	58.53 ± 11.15	*p* = 0.01 (0.68)	1.56 (0.34/3.03)	51.92 ± 11.36	51.87 ± 10.62	*p* = 0.32 (0.53)	−0.05 (−0.97/0.85)	*p* = 0.99 (0.02)	*p* = 0.28	*p* = 0.94 (0.04)	*p* = 0.25
DRY LEAN (kg)	15.88 ± 5.11	15.73 ± 5.06	*p* = 0.29 (0.28)	−0.14 (−0.47/0.13)	12.71 ± 4.75	11.91 ± 4.88	*p* = 0.89 (0.28)	−0.14 (−2.60/0.43)	*p* = 0.71 (0.05)	*p* = 0.18	*p* = 0.58 (0.09)	*p* = 0.01
PhA (^0^)	4.94 ± 0.77	5.60 ± 1.96	*p* = 0.04 (0.42)	0.65 (−0.33/1.14)	5.29 ± 2.18	5.18 ± 2.41	*p* = 0.14 (0.28)	−0.10 (−1.85/0.42)	*p* = 0.21 (0.10)	*p* = 0.80	*p* = 0.22 (0.14)	*p* = 0.03

Note: ES: effect size; *p*: *p*-value; Δ: average difference; CI: 95% confidence interval (lower bound/upper bound); BF: body fat; BCM: body cell mass; BFMI: body fat mass index; FFMI: fat-free mass index; BMR/BW: ratio of basal metabolic rate and body weight; BMR: basal metabolic rate; TBW: total body water; ECW: extracellular water; ICW: intracellular water; Pha: phase angle; Data are expressed as mean ± SD; *p* < 0.05: baseline vs. 6 months follow-up; *p* < 0.05: group A vs. B.

## Data Availability

The data presented in this study are available on request from the corresponding author. The data are not publicly available due to ethical restrictions.

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
