# Peer review of "Effects of a Combined Intradialytic Exercise Training Program on Functional Capacity and Body Composition in Kidney Transplant Candidates"

_jfmk, 2023, doi:10.3390/jfmk8010009_

Round 1

Reviewer 1 Report

Thank you for submitting your valuable manuscript to this journal.

The aim of the present study was to evaluate the effects of a 4-month combined intradialytic exercise training on functional capacity and body composition indices in KT candidates.

This is a very well-designed randomized clinical trial with proper procedures and a comprehensive explanation of the materials and methods. Although the number of participants is small, significant differences in some results can make it borderline acceptable.

Another advantage of the study protocol is its duration of intervention which is a 4-month exercise program suitable to make the changes on functional capacity and body composition parameters.

There are some points that need to be considered in the paper:

Major points:

1-      How do you justify the reverse results obtained from 10-STS test? As it is shown in the table and mentioned in the results, the time needed for performing 10 times sit to stand is significantly increased in the intervention group and on the other hand the control group had significantly better performance with shorter time on this test!

2-      Since none of the functional capacity parameters were significantly different between the groups, how did you concluded that exercise intervention improved physical function in HD patients compared with the usual care?
It seems that you have just to conclude that exercise can improve some physical function parameters (handgrip strength and 6MWD) in the intervention group without any significant difference with the usual care.

3-      In table 1, you have mentioned some baseline characteristics of the groups without any significant difference among them. But in table 3, there is a significant difference in baseline BF (Kg) between groups (21.61 vs. 28.57, P=0.05)! It is more strange that the post intervention measures are not significantly different between groups (with P=0.06) but with a wider numerical difference (20.52 vs. 28.82).

4-      Line 179; How did you calculate body fat mass index (BFMI) and Fat Free Mass Index (FFMI)?
Did you divide the body fat mass or fat free mass to body weight or to body surface area? Please mention the method.

Minor points:

1-      Some BIA parameters such as Resistance (Ro), Reactance (Xc), Impedance of 5khz (Z5) and Impedance of 50khz (Z50) have not independent clinical value and are used to calculate other important parameters in BIA, so I would like to suggest you to remove them from the study parameters.

2-      It is necessary to define the variable (HD vintage), and to mention the abbreviations (URR and Kt/V) in table 1 footnote.

3-      Line 239, 240 and 241; please remove the repeated data about HGS, 10-STS and 6MWD that are mentioned in table 2.

According to the major points mentioned above, the conclusion of the manuscript needs to be revised.

Good luck.

Reviewer 2 Report

Dear Authors,

You have written an interesting paper. However, some parts need to be addressed for greater clarity.

The introduction is clear and nicely leads to the main rationale.

Methods

How was your sample size determined (G*Power or any other method)? Report

What was the participants' Physical activity history before the study? Report as this can affect your results. If you don't have this info add this in the limitations section.

Six-minute walk test - report the exact model of all equipment used from time to BP and HR. Report what value was taken into further analysis. Was there any warmup before the test? report

Line 152 - chair about 45cm high / how do you mean about. What was the model of the chair? Was it adjusted for every participant? What was the knee angle? Report

How many times was the sit-to-stand test done? only once

Hand grip strength - the first 2 sentences need to be in the introduction and not here in the methods. Amend

the instructions are confusing - first you state the hand was extended sideways / or do you mean rotated-turned? Also, the 10s break is too short! This needs to be added in the limitations and/or backed up by references! How long were these contractions - what were the instructions? Report

Report how were weight and height measured.

BIA analysis - was it done on a non-dialysis day, what were the instructions 1 day before the measurement? report

The exercise program is poorly described. I recommend adding a flowchart or a graph with pictures for a better presentation and especially reproducibility. Please amend

Fig 1 needs to be of better quality as it is hard to read. Amend

Report effect size in ANOVA and t-tests!

The discussion should also mention the limitations of BIA measurements in dialysis patients. I recommend the following paper https://www.mdpi.com/2073-8994/13/1/150

Limitations of the study should be extended as they are really modest.

Overall an interesting and promising study that needs some more work.

Round 2

Reviewer 1 Report

Thank you for the revision of your valuable manuscript.

For the comment about the 10-STS test, it is mentioned that the results of this study did not show a significant improvement either within group A, or between groups, but as it is evident in table 2, the result has significantly deteriorated in the intervention group (A) (22.4 to 24.8 Sec, P<0.05) and has significantly improved in the control group (B) (22.07 to 20.42 Sec, P<0.05) with a significant between group difference (P=0.05). This is the opposite of your conclusion in the paper.

For 6-MWD, the previous version of the paper didn’t show a significant difference between groups, but now it is mentioned that analysis on the average differences within and between groups shows a significant difference (p=0.04). As you can see in the table 2, 6-MWD for the control group (B) (420.55 to 408.54 Meters, P=0.16) didn’t significantly change before and after the study process, with the average distance (12.01 Meters) and the confidence interval (7.98 / 16.04). Since zero is not included in the confidence interval, the difference is significant, but the reported P value is 0.16 which doesn’t show a significant difference! It seems that there are important errors in the new analyses of the study!

Good luck.

Reviewer 2 Report

Dear Authors

Thank you for fully addressing my comments. In my opinion, the quality and clarity of the paper improved. Therefore, I recommend acceptance.

Kind regards

Round 3

Reviewer 1 Report

Thank you for the revision of your valuable manuscript.

For the comment about the 10-STS test, it is true that the result did not show a significant improvement, on the contrary, it has significantly deteriorated in the intervention group and significantly improved in the control group. This reverse result needs to be discussed in the discussion section. 

The comment on the confidence interval of 6-MWD test, needs an explanation from a statistics journal and not a psychological journal!

Good luck.

Round 4

Reviewer 1 Report

Thank you for the revision of your valuable manuscript. 

Despite some drawbacks in the statistical analysis and other limitations of the study, given that the strengths and weaknesses are mentioned in the discussion, the paper is acceptable for publication.

Good luck.